# The cardiovascular polypill as baseline treatment improves lipid profile and blood pressure regardless of body mass index in patients with cardiovascular disease. The Bacus study

**José Alejandro Chávez Fernández**[1]*, **Marcelo Ramírez Mendoza**[2], **Hermelinda Kassck Ipinaa**[2], **Luís Antonio Sánchez Ángeles**[2], **Antonio González Chávez**[3], **Galileo Escobedo**[4], **Lucía Angélica Méndez-García**[4]

1 Clínica de Enfermedad Coronaria e Insuficiencia Cardiaca, Servicio de Cardiología, Hospital General de México, Dr. Eduardo Liceaga, Mexico City, Mexico, 2 Servicio de Cardiología, Hospital General de México, Dr. Eduardo Liceaga, Mexico City, Mexico, 3 Servicio de Medicina Interna, Hospital General de México, Dr. Eduardo Liceaga, Mexico City, Mexico, 4 Departamento de Proteómica y Metabolismo, Hospital General de México, Dr. Eduardo Liceaga, Mexico City, Mexico

* dralejandrochavez.20@gmail.com

## Abstract

### Background

Pharmacological treatment with lipid-lowering and antihypertensive drugs has been proposed as a strategy to improve excess cardiovascular (CV) risk among obese individuals. The present study aimed to assess whether the CV polypill (Sincronium®) could be an effective strategy to help improve CV risk factor control in obese/overweight individuals requiring secondary prevention.

### Methods

This was an observational, retrospective study reviewing the hospital medical records of 479 patients with established CV disease who initiated treatment with the CV polypill between 2013 and 2019 at a general hospital in Mexico. Patients were grouped as normal weight, overweight or obese according to their initial body mass index (BMI). We collected blood pressure (BP), lipid profile, and vascular age at the last visit recorded during the period following treatment.

### Results

At the end of the study, all assessed lipid parameters improved compared to baseline regardless of the initial BMI category (all p<0.001). There was an increase from baseline regarding the proportion of patients with at target low-density lipoprotein cholesterol after treatment (2.3% vs. 30.1%; p<0.001), more than 80% of patients achieved triglyceride levels <200 mg/dL (p<0.001), and more than 80% achieved target BP levels in all BMI subgroups

**Data Availability Statement:** Data cannot be shared publicly because of the Hospital General de México policy. Data are available from the Education and Research department of the Hospital General de México for researchers who meet the criteria for access to confidential data by contacting Dr. Martín Gómez Silva via email (mogones.cdmx@gmail.com).

**Funding:** The study was partly funded by Ferrer Internacional SA. The funder had no role in study design, data collection, data interpretation, or decision to submit it for publication.

**Competing interests:** JACF received payment from Ferrer Internacional SA to conduct the data analysis and prepare the manuscript. This does not alter our adherence to PLOS ONE policies on sharing data and materials. The other authors report no relationships that could be construed as a conflict of interest.

(p<0.001). The subanalyses in the elderly population yielded similar results, with a significant overall improvement in lipid and BP control after initiating the CV polypill strategy.

## Conclusions

The use of the CV polypill as baseline therapy for secondary prevention seems to be a reasonable strategy that enhances CV risk factor control regardless of the patient's BMI.

## Introduction

Overweight and obesity are major public health issues with epidemic proportions globally and a prevalence that has constantly been increasing during the past 30 years in most countries [1]. In 2016, the World Health Organization (WHO) estimated that 39% (38% of men and 40% of women) of the adult population worldwide was overweight and 13% (11% of men and 15% of women) was obese [1]. Furthermore, it is estimated that 2.8 million people die each year as a result of being overweight or obese [1]. The burden of obesity and being overweight is of particular concern in Latin America (LATAM), where the average increase in body mass index (BMI) has risen twice as fast as the average global increase [2]. Among LATAM countries, Mexico has been recently highlighted as one of the world's nations with the highest rates of adults being overweight (72.5%) and obese (33%), and it is the country with the largest predicted reductions in life expectancy due to overweight-related diseases (4.2 years) for the period from 2020 to 2050 [3].

There is strong evidence that excess body weight increases the risk and incidence of cardiovascular diseases (CVDs) through associated metabolic risk factors such as hypertension, dyslipidemia and hyperlipidemia, or type 2 diabetes mellitus (T2DM) [4–7]. For instance, CVDs cause nearly 70% of deaths in subjects with a high BMI and more than 60% among obese subjects [8]. In particular, blood pressure, cholesterol, and glucose account for 46% of the excess coronary heart disease (CHD) risk mediated by being overweight and obese and 76% of the excess stroke risk, with blood pressure as the most important mediator (31% and 65% of excess risk of CHD and stroke, respectively) [9]. Moreover, traditional weight-loss interventions (including dietary/lifestyle change, weight loss medications, or surgery), unless sustained and long-term, have modest and inconsistent benefits in obese individuals with cardiovascular (CV) risk factors [10]. For this reason, some authors advocate that additional pharmacological treatment with lipid-lowering and antihypertensive drugs is needed in obese individuals with CV risk to attain clinically significant improvements in risk factors [9, 10].

It is widely recognized that there is a continuous, graded, and strong relationship between CV risk factors (hypertension, dyslipidemia, and T2DM) and the risk of premature death from CVD [11]. Since multiple risk factors usually cluster in individuals, the need for a multi-targeted treatment approach based on combination therapy with two or more drugs is the norm rather than the exception. However, complex medication regimens result in poor therapeutic compliance and persistence, which are known to be linked to increased risk of recurrent or new CV events and poor clinical outcomes [12]. With the emergence of the CV polypill, a single tablet in general composed of a minimum of a statin, an antiplatelet drug, and an antihypertensive agent, these drawbacks can now be handled more easily [13]. The CV polypill strategy has also been shown to be cost-effective [13–17] and is currently mentioned in several clinical guidelines [18–20]. However, most of the available studies implementing CV polypills were prospective randomized trials (RCTs) using placebo as a comparator in patients in

primary prevention, with limited data on its impact on clinical outcomes in the secondary prevention setting [21–23]. Moreover, secondary prevention data arose from studies mainly conducted in European populations [24, 25].

Based on the suggestion to pharmacologically target metabolic mediators in obese/overweight subjects with CV risk factors, we assessed whether the CV polypill could be an effective strategy to improve the associated comorbidities and excess CV risk in these individuals when on secondary prevention. For this, we conducted a real-world observational study in Mexico, where obesity has been declared an epidemiological emergency because, together with its alarming high prevalence, it remains under-diagnosed, under-treated and, when treated, traditional lifestyle interventions are not effective [26].

## Methods

### Study design and data source

This was an observational, single-center, retrospective, physician-led medical record review study conducted in the General Hospital of México (Mexico City, Mexico) Coronary Unit. The medical cardiologists responsible for patients requiring secondary CVD prevention treatment because of established coronary heart disease (CHD) retrospectively reviewed the medical records of consecutive eligible patients receiving the CV polypill between January 2013 and January 2019.

Because of the study's retrospective nature on deidentified data, informed consent was waived. Ethical approval for this study was obtained from the General Hospital of México Ethics Committee. The study was conducted in accordance with the Declaration of Helsinki (2013).

### Study population

All adult patients of both sexes attending the Coronary Unit from January 2013 to January 2019 had their medical records queried via the hospital's electronic health record system. Of the eligible medical records, we identified in chronological order those from patients with a diagnosis of established CHD (i.e., unstable CHD or acute coronary syndrome [myocardial infarction and unstable angina]) requiring secondary CVD prevention treatment. Among them, we selected those who switched their usual treatment to any formulation of the CV polypill (acetylsalicylic acid 100 mg, simvastatin 40 mg and ramipril 5 or 10 mg) at their first visit to the Unit regardless of whether the previous treatment was stable or who initiated treatment with the CV polypill for the first time because they were not taking one or more of the separate individual components of the three drug classes as recommended by the guidelines [18, 27]. Of these selected patients, only those who had data from at least one follow-up visit after initiating treatment were included. For subjects with hypertension fulfilling the above criteria but under suboptimal control, additional antihypertensive agents were prescribed in addition to the CV polypill. Subjects were excluded if the medical records did not include laboratory results or had no BP measurement at their initial and follow-up visit, or the values dissented from the diagnosis or were unreadable.

### Data collection and study measures

The data collected were part of the general cardiological assessments and management of the patients per routine clinical practice and national/local protocols and guidelines. Baseline characteristics, exposure, and outcomes were defined a priori to reduce the bias inherent to retrospective chart review studies. The qualitative and quantitative data were integrated into

an Excel spreadsheet and deidentified from the electronic chart by removing the name, date of birth, and day of visit.

At baseline, defined as the date when the patient initiated treatment with the CV polypill, the following variables were collected: age, gender, history of CV events, history of T2DM, smoking status, weight and height to calculate BMI, systolic and diastolic BP (SBP and DBP, respectively, as determined in the office setting), total cholesterol (TC), low-density lipoprotein cholesterol (LDL-c), high-density lipoprotein cholesterol (HDL-c), and triglycerides (TG). With the obtained data, we calculated the vascular age, which is defined as the CVD risk of an individual -derived from the Framingham risk tables and SCORE scale equations- transformed to the age of a person with the same risk but controlled risk factors [28].

The baseline BP and laboratory values used in the analyses corresponded to the last available measurement immediately before CV polypill initiation. The same variables were extracted from the last available visit recorded during the period following treatment with the CV polypill (end of study). Regarding pharmacological treatment, we extracted data on medications (antihypertensive drugs, lipid-lowering drugs, glucose-lowering drugs, antiplatelet agents, anticoagulants, and antiarrhythmic drugs) taken prior to the use of the CV polypill, concomitant to the CV polypill during follow-up, and at the end of the study. Adverse events spontaneously reported by the patients during CV polypill therapy were also recorded.

## Statistical analysis

Patients were stratified according to their baseline BMI values into the three following WHO categories [29]: a) normal weight if BMI was between 18.5 and 24.9 kg/m$^2$; b) overweight if BMI was between 25 and 29.9 kg/m$^2$; and c) obesity defined as a BMI of $\geq$30 kg/m$^2$. Descriptive analyses were summarized by mean and standard deviation (SD) for continuous variables and absolute frequency and percentages for categorical variables. The effectiveness of the CV polypill was assessed through the difference between baseline and the last follow-up visit in the mean value of the lipid profile (TC, HDL-c, LDL-c, and TG), lipid ratios (i.e., TC/HDL-c and LDL-c/HDL-c), blood pressure (i.e., SBP and DBP), and vascular age; the statistical significance of the change was evaluated by a Student's $t$-test for repeated measures. Moreover, the change in the proportion of patients with on-target LDL-c levels (i.e., <70 mg/dL) and patients with normal or borderline high TG levels (<150 and <200 mg/dL, respectively) were compared to baseline between groups by a Chi-square test. Please note that the target levels for LDL-c and recommended levels for TG were those of the 2016 European Society of Cardiology (ESC) guidelines on lipid management [30], since this study was performed before the latest 2019 ESC guidelines were published [31]. Additionally, the difference in the proportions of patients who achieved the target BP (i.e., <140/90 mmHg or <130/85 mmHg for patients with T2DM) between baseline and after the use of the CV polypill and, among them, the percentage that needed concomitant medication or not was also assessed through the Chi-square test. A sub-analysis was conducted in the elderly population where all outcomes were assessed in subjects over 65 years. P values were two-tailed, and $p$<0.05 was considered statistically significant. All the statistical analyses were performed using the statistical package SPSS 22.0 (SPSS Inc., Chicago, Illinois).

## Results

Out of 513 eligible patients who were previously treated with the standard of care (i.e., multipill strategy), 479 with an established CHD and complete medical records who switched to or initiated treatment with the CV polypill were included in the study. Most of them were overweight or obese (45% [n = 216] and 38.2% [n = 138], respectively) and 16.7% (n = 80) were at normal weight. The baseline characteristics of the population are shown in **Table 1**. The mean

**Table 1. Baseline demographic and clinical characteristics of patients.**

| Characteristic | All (N = 479) | Normal weight (n = 80) | Overweight (n = 216) | Obese (n = 183) | P-value |
|---|---|---|---|---|---|
| **Age (years), mean (SD)** | 63.9 (12.4) | 67.7 (11.7) | 63.8 (12.6) | 62.3 (12.3) | 0.005 |
| **Gender (male), n (%)** | 306 (63.8) | 47 (58.7) | 109 (50.4) | 150 (81.9) | <0.001 |
| BMI (kg/m$^2$), mean (SD) | 29.1 (4.7) | 23.1 (1.4) | 27.4 (1.3) | 33.7 (4.04) | <0.001 |
| **Cardiovascular risk factors, n (%)** | | | | | |
| Hypertension | 287 (59.9) | 47 (58.7) | 128 (59.2) | 112 (61.2) | 0.900 |
| *Time since diagnosis (years), mean (SD)* | 9.4 (8.6) | 7.8 (7.9) | 9.7 (8.7) | 9.7 (8.9) | 0.421 |
| Hypercholesterolemia | 75 (15.7) | 13 (16.2) | 34 (15.7) | 28 (15.3) | 0.868 |
| Hypertriglyceridemia | 392 (81.8) | 65 (81.2) | 180 (83.3) | 147 (80.3) | 0.732 |
| Diabetes Mellitus | 203 (42.4) | 31 (38.7) | 92 (42.1) | 80 (43.7) | 0.752 |
| *Time since diagnosis (years), mean (SD)* | 10.8 (7.9) | 13.6 (8.5) | 9.7 (7.1) | 11 (8.3) | 0.061 |
| Smoker | 291 (60.8) | 44 (55) | 131 (60.6) | 116 (63.3) | 0.439 |
| **Cardiovascular disease, n (%)** | | | | | |
| Myocardial infarction | 458 (95.6) | 75 (93.7) | 210 (97.2) | 173 (94.5) | 0.028 |
| Unstable angina | 14 (2.9) | 3 (3.8) | 6 (2.7) | 5 (2.7) | 0.891 |
| Other* | 7 (1.5) | 2 (2.5) | 2 (0.9) | 3 (1.6) | 0.585 |
| **Cardiovascular risk score** | | | | | |
| Vascular age (years), mean (SD) | 71.4 (11.2) | 73.9 (8.6) | 71.3 (11.0) | 70.5 (12.3) | 0.082 |
| **Blood pressure (mmHg), mean (SD)** | | | | | |
| *SBP* | 133.2 (17.6) | 133.1 (18.3) | 132.1 (17.3) | 134.6 (17.6) | 0.371 |
| *DBP* | 76.1 (10.9) | 75.6 (12.3) | 74.8 (10.1) | 77.7 (11.0) | 0.82 |
| **Lipid profile (mg/dL), mean (SD)** | | | | | |
| *TC* | 228.7 (42.9) | 228.3 (42.8) | 229.6 (43.6) | 227.9 (42.3) | 0.922 |
| *TG* | 247.0 (120.7) | 257.8 (158.3) | 246.8 (117.4) | 242.4 (105.1) | 0.638 |
| *LDL-c* | 132.0 (33.7) | 127.3 (32.6) | 132.7 (34.2) | 133.3 (33.6) | 0.381 |
| *HDL-c* | 38.7 (10.4) | 39.0 (9.4) | 37.9 (10.1) | 39.5 (11.1) | 0.287 |
| **Baseline medications concomitant to the CV polypill, n (%)** | | | | | |
| *CCBs* | 193 (40.3) | 35 (43.8) | 75 (34.7) | 83 (45.4) | 0.077 |
| *Diuretics* | 25 (5.2) | 5 (6.3) | 10 (4.6) | 10 (4.6) | 0.841 |
| *Beta-blockers* | 229 (47.8) | 37 (46.3) | 112 (51.9) | 80 (43.7) | 0.257 |
| *Non-statin lipid lowering drugs* | 8 (1.7) | 3 (3.8) | 2 (0.9) | 3 (1.6) | 0.178 |
| *Antiplatelet agents (non-ASA)* | 74 (15.4) | 11 (13.8) | 38 (17.6) | 25 (13.7) | 0.501 |
| *Glucose-lowering drugs* | 30 (6.3) | 4 (5.0) | 15 (6.9) | 11 (6.0) | 0.816 |
| *Anticoagulants* | 3 (0.6) | 0 (0.0) | 2 (0.9) | 1 (0.5) | 1 |
| *Antiarrhythmics* | 12 (2.5) | 1 (1.3) | 6 (2.8) | 5 (2.7) | 0.857 |
| Time on treatment (months), *mean (SD)* | 56.64 (18.7) | 54.21 (16.2) | 55.90 (15.6) | 58.60 (22.7) | 0.169 |

*Other CVDs include stroke, peripheral vascular disease or hypertensive heart disease

ASA: acetylsalicylic acid; BMI: body mass index; CCB: calcium channel blocker; CV: cardiovascular; CVD: cardiovascular disease; DBP: diastolic blood pressure; HDL-c: high-density lipoprotein cholesterol; LDL-c, low-density lipoprotein cholesterol; SBP: systolic blood pressure; SD: standard deviation; TC: total cholesterol; TG: triglycerides:

time of treatment with the CV polypill was 8.1 years (SD = 3.4) without differences between subgroups. Overall, the mean age of the population was 63.9 years (SD = 12.4), more frequently male (63.8%), and the mean BMI was 29.1 kg/m$^2$ (SD = 4.7). Sixty percent of patients had hypertension, 81% hypertriglyceridemia, the mean LDL-c value was 132.0 mg/dL (SD = 33.7), and the mean vascular age was 7.5 years older than the mean biological age.

Based on BMI categories, there were no significant differences between normal weight, overweight, and obese individuals except for age, with subjects with normal weight older than those in the other two groups (67.7 years vs. 63.8 and 62.3 years in overweight and obese individuals, respectively; p = 0.005), sex, with more male patients in the obese group than in the other groups (81.9% vs 58.7% and 50.4% in those with normal weight and overweight, respectively; p<0.001), and the history of myocardial infarction, which was slightly less frequent among subjects with normal weight (93.7% vs. 97.2% of those overweight and 94.5% of obese subjects; p = 0.028).

Prior to initiating or switching to the CV polypill strategy, 40.7% of patients were on treatment with acetylsalicylic acid (ASA), 45.7% with statins, and 11.7% with renin-angiotensin-aldosterone system (RAAS) inhibitors (S1 Fig). No significant differences were found between groups regarding prior use of ASA and statins, but RAAS inhibitors were more frequently prescribed to normal weight and overweight patients than to obese patients (p = 0.017) (S1 Fig).

In normal weight patients, the previously prescribed statins were mostly equipotent to the 40 mg simvastatin of the CV polypill (58.1% of cases), while in overweight and obese subjects almost half of them were on more potent statins (50.6% and 53.7%, respectively) (S2 Fig). In contrast, the use of less potent prior statins was uncommon in all groups (16%, 4.4%, and 6.1% of cases of normal weight, overweight, and obese patients, respectively). Regarding RAAS inhibitors, agents less potent than the 5 mg or 10 mg ramipril contained in the CV polypill were more frequent than equipotent drugs only in normal weight patients (69.2% vs. 30.8%), whereas the prior use of equipotent drugs was more frequent than less potent drugs in overweight and obese patients (58.3% vs. 33.3% and 45.5% vs 36.4%, respectively; S2 Fig).

## Treatment discontinuations and concomitant medications

During the study period, 66 patients (13.8%) interrupted the treatment with the CV polypill (S1 Fig). In most cases (96.4%), the patients were taking the 10 mg ramipril dose. The proportion of patients who interrupted the treatment was similar between BMI groups (15%, 12.9%, and 14.2% in the normal weight, overweight, and obese groups, respectively; p = 0.081). Moreover, none of the reported reasons for CV polypill discontinuation was significantly different between BMI groups (S1 Fig). Among those who interrupted the treatment, the most frequent reason was cough (21 patients; 31.8%), which occurred in a similar number of cases in each BMI group (25%, 32.1%, and 34.6% in the normal weight, overweight, and obese groups, respectively; p = 0.886; S1 Table). Hypotension was reported as the cause for discontinuation in 4 patients (6.1%), economic reasons in 3 patients (4.5%), and atrial fibrillation, severe heart failure, bleeding, and pain occurred in 2 patients each (3.0%). All other reasons were observed in only one patient (1.5%), with lack of efficacy reported in one obese subject (S1 Table). None of the reported reasons for CV polypill discontinuation was significantly different between BMI groups (S1 Table).

The summary of the concomitant drugs that patients needed to control CV risk factors or comorbid diseases besides the CV polypill baseline treatment at the end of the follow-up is shown in Table 2. Overall, the most frequently required additional medications were antihypertensive drugs (94.4% of patients), followed by glucose-lowering drugs and antiplatelet agents (8.1% and 5.2%, respectively). In contrast, lipid-lowering medications, anticoagulants, and antiarrhythmics were prescribed to less han 2.1% of the patients (0.4%, 0%, and 2.1%, respectively). None of the patients received additional RAAS inhibitors nor an additional dose of simvastatin or other statins. The overall number of patients requiring concomitant medications did not differ between study periods (i.e., before switching to the CV polypill, at baseline, and the end of follow-up) except for non-ASA antiplatelet agents, which were less frequently

**Table 2. Concomitant treatments to cardiovascular polypill at the end of follow-up.**

| Treatment | All (N = 479) | Normal weight (n = 80) | Overweight (n = 216) | Obese (n = 183) | P-value |
|---|---|---|---|---|---|
| **Antihypertensive drugs, n (%)** | 452 (94.4) | 76 (95.0) | 207 (95.8) | 169 (92.3) | 0.298 |
| *Calcium channel blockers* | 202 (42.2) | 38 (47.5) | 81 (37.5) | 83 (45.35) | 0.163 |
| *Diuretics* | 32 (6.7) | 6 (7.5) | 16 (7.4) | 10 (5.5) | 0.704 |
| *Beta blockers* | 218 (45.5) | 32 (40.0) | 110 (50.9) | 76 (41.5) | 0.095 |
| *RAAS inhibitors* | 0 (0.0) | 0 (0.0) | 0 (0.0) | 0 (0.0) | - |
| **Lipid-lowering drugs, n (%)** | 2 (0.4) | 1 (1.3) | 1 (0.5) | 0 (0.0) | 0.452 |
| *Fibrates* | 2 (0.41 | 1 (1.3) | 1 (0.5) | 0 | 0.452 |
| *Ezetimibe* | 0 (0.0) | 0 (0.0) | 0 (0.0) | 0 (0.0) | - |
| *Other (PSK9i, Omega 3FA)* | 0 (0.0) | 0 (0.0) | 0 (0.0) | 0 (0.0) | - |
| *Additional simvastatin dose* | 0 (0.0) | 0 (0.0) | 0 (0.0) | 0 (0.0) | - |
| *Other statins* | 0 (0.0) | 0 (0.0) | 0 (0.0) | 0 (0.0) | - |
| **Antiplatelet agents (non-ASA), n (%)** | 25 (5.2) | 0 | 0 | 0 | - |
| *Clopidogrel* | 11 (2.3) | 0 | 5 (2.3) | 6 (3.3) | 0.305 |
| *Ticagrelor* | 14 (2.9) | 3 (3.8) | 5 (2.3) | 6 (3.3) | 0.770 |
| *FDC with antiplatelet* | 0 (0.0) | 0 (0.0) | 0 (0.0) | 0 (0.0) | - |
| **Glucose-lowering drugs, n (%)** | 39 (8.1) | 6 (7.5) | 20 (9.3) | 13 (7.1) | 0.716 |
| *Oral antidiabetics* | 38 (7.9) | 6 (7.5) | 19 (8.8) | 13 (7.1) | 0.813 |
| *FDC of antidiabetic drugs* | 57 (11.9) | 6 (7.5) | 28 (13.0) | 23 (12.6) | 0.409 |
| *Insulin* | 2 (0.4) | 0 (0.0) | 2 (0.9) | 0 (0.0) | 0.655 |
| **Anticoagulants, n (%)** | 0 (0.0) | 0 (0.0) | 0 (0.0) | 0 (0.0) | - |
| **Antiarrhythmics, n (%)** | 10 (2.1) | 1 (1.3) | 5 (2.3) | 4 (2.2) | 0.844 |

ASA: acetylsalicylic acid; FDC: fixed-dose combination; RAAS: renin-angiotensin-aldosterone system

required at the end of the study (p<0.0001; **S3 Fig**). Finally, the number of patients needing concomitant therapies was similar between BMI subgroups for all assessed drug classes except for lipid-lowering drugs, which were more frequently added to normal-weight patients at baseline (**S2 Table**).

## Effects of the CV polypill on lipid parameters, blood pressure, and CV risk

After initiating the CV polypill treatment, there were statistically significant improvements compared to baseline in all assessed lipid parameters regardless of the baseline BMI category (all p<0.001) (**Fig 1**). Overall, the decrease in lipid parameters (TC, TG, and LDL-c) after treatment exceeded 30% in all cases, and the minimum increase in HDL-c was 13% (**Fig 1**). In particular, the decrease in TC ranged from 31% to 33% across BMI groups (**Fig 1A**), the reduction in TG levels was higher among normal weight subjects than among the other groups (−43% vs. −32–33%; **Fig 1B**), the reduction in LDL-c levels was almost identical in all weight groups (−37–38%; **Fig 1C**), and the increase in HDL-c levels was also similar between groups (13–18%; **Fig 1D**). Regarding the TC/HDL-c lipid ratio (Castelli's risk index-I), we observed a larger decrease among normal weight and overweight patients (−41% and −45%, respectively) than in obese individuals (−39%; **Fig 1E**). Finally, the LDL-c/HDL-c ratio (Castelli's risk index-II or atherogenic index) almost halved the baseline value, and the decrease was similar across weight groups, although slightly higher among overweight and obese participants (−50% and −49%, respectively) than in normal weight individuals (−47%; **Fig 1F**).

Regarding BP values, there was a significant decrease in all BMI groups compared to baseline (p<0.001; **Fig 2**), and the magnitude of the reduction was between 10% and 11% for both

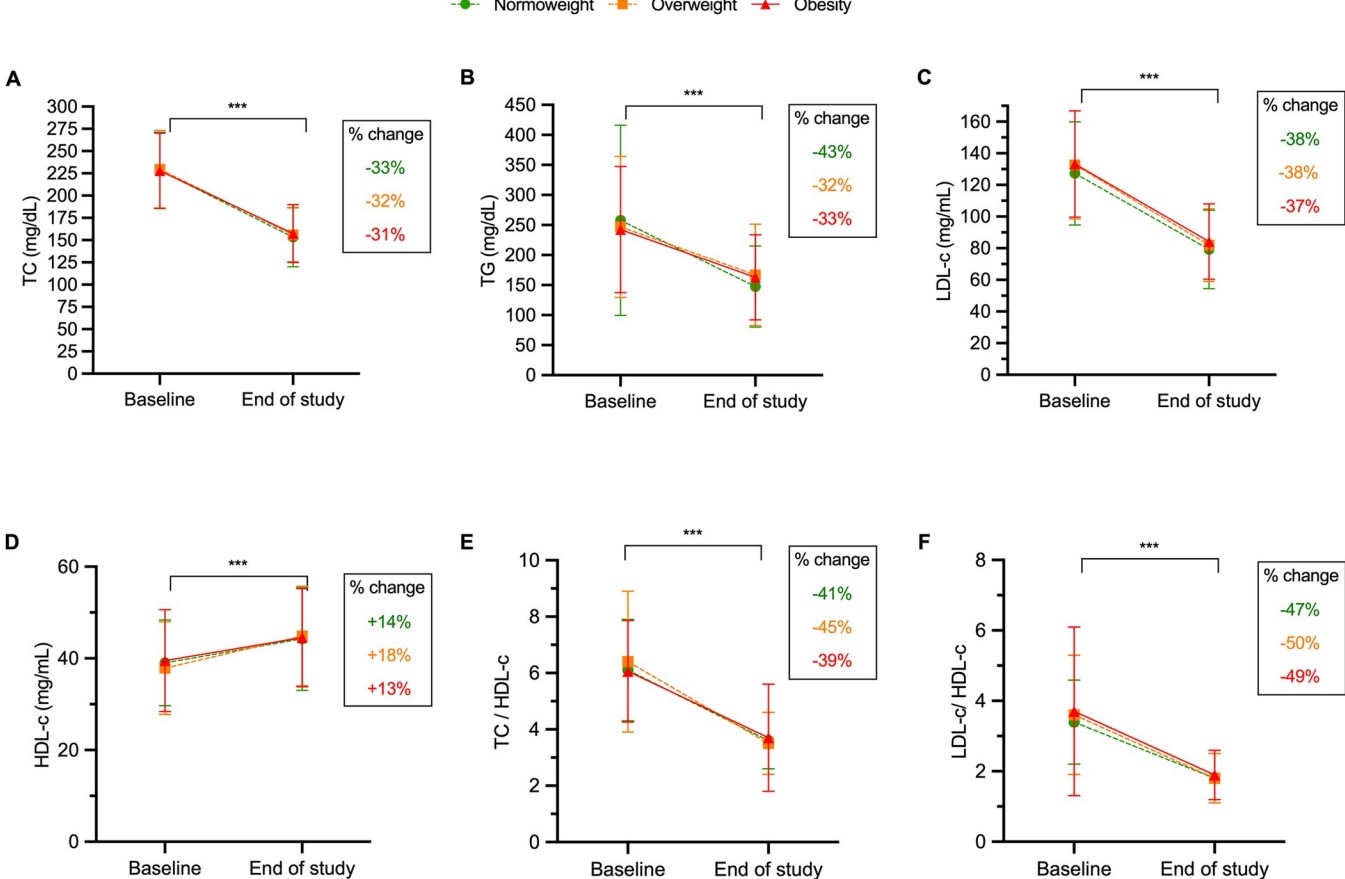

**Fig 1. Change in levels and additional change (%) in lipid parameters with the CV polypill compared to baseline.** ***P<0.001 in all weight groups. HDL-c: high-density lipoprotein cholesterol; LDL-c: low-density lipoprotein cholesterol; TC: total cholesterol; TG: triglycerides.

SBP and DBP across weight groups (**Fig 2A and 2B**). Regarding CV risk, we observed that vascular age decreased by 6.6 years in the normal weight group and about 5.2 and 5.0 years in the overweight and obese groups, respectively (**Fig 2C**).

## Control of lipid parameters and blood pressure during the CV polypill treatment period

The proportion of patients attaining target levels of lipid parameters (LDL-c and TG) and BP at the end of follow-up was significantly higher after treatment with the CV polypill than at baseline ($p<0.001$) (**Fig 3A and 3B**). In the case of LDL-c, the proportion of patients with target levels recommended for patients with established CVD (i.e., <70 mg/dL) was very low at baseline (2.3% overall) but increased significantly to 30.1% after treatment with the CV polypill, with small variations according to BMI categories (28.7% to 33.8% across groups) (**Fig 3A**). The improvement was much more evident regarding recommended TG levels, as less than 40% of patients (32.8% to 36.6%) were at <200 mg/dL before CV polypill initiation, but more than 80% of patients achieved these levels in all BMI subgroups (80.1–82.0%) after treatment initiation (**Fig 3A**). Similarly, the proportion of patients that were able to attain stricter TG levels <150 mg/dL almost doubled after treatment with the CV polypill (16.7% to 19.7% at baseline and 47.2% to 56.3% after treatment across BMI groups) (**Fig 3A**).

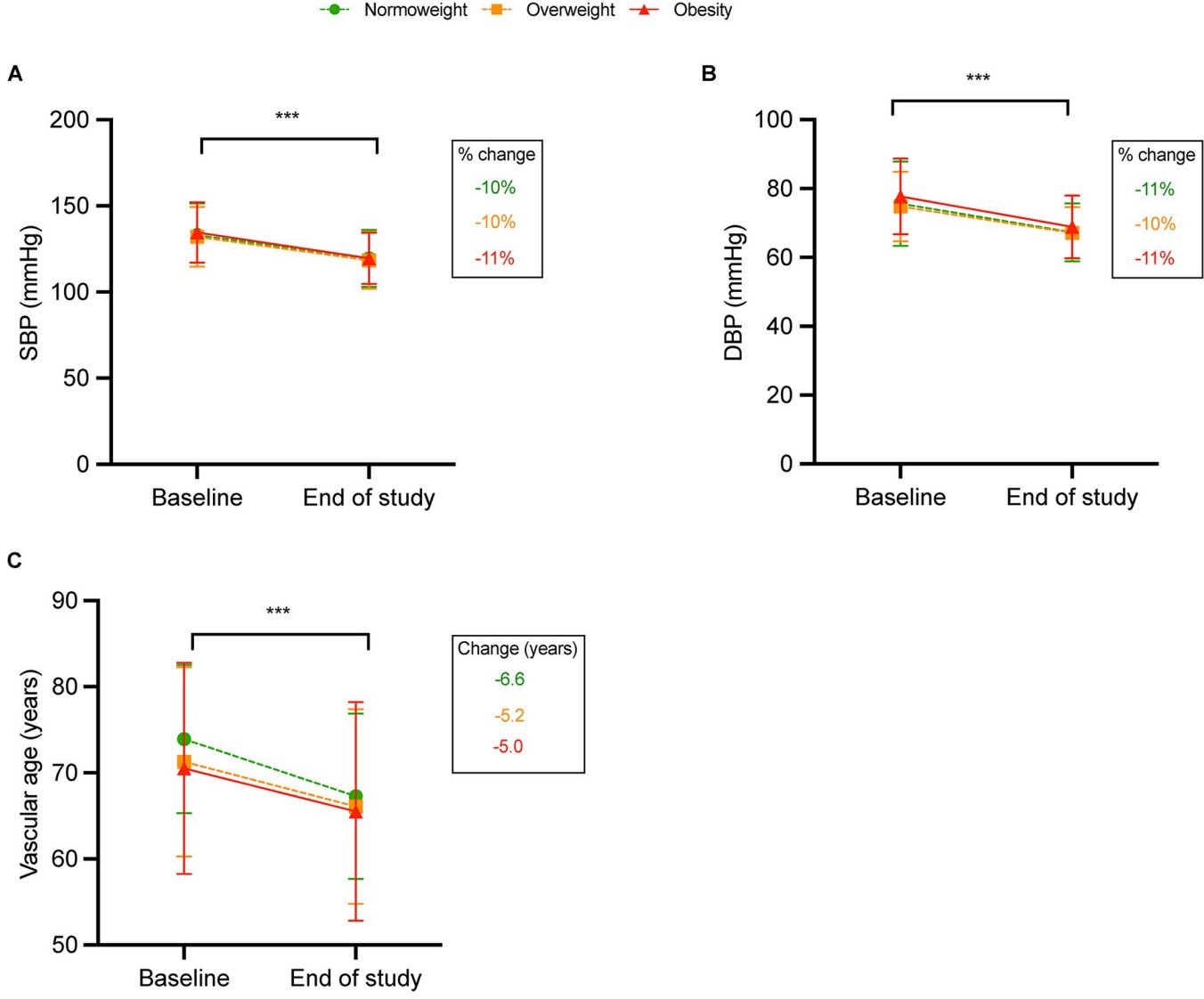

**Fig 2.** Change in levels and additional change (%) in blood pressure (A and B), vascular age (C) with the CV polypill compared to baseline. ***P<0.001. DBP: diastolic blood pressure; SBP: systolic blood pressure.

The improvement was also noticeable regarding BP, with about half of the patients at the recommended BP levels at baseline (45.9% to 52.5% across BMI groups) but above 80% after CV polypill treatment (84.7% to 88.4% across BMI groups) (**Fig 3B**). Also, about 2/3 of at-target patients needed concomitant BP-lowering drugs to remain controlled besides the baseline CV polypill treatment, and the remaining 1/3 were controlled with the CV polypill as monotherapy (**Fig 3B**).

## Outcomes in the elderly subpopulation

Out of the total population, 224 patients (46.8%) were over 65 years, with 80% of them overweight or obese (**S3 Table**). There were no significant differences across BMI groups except that subjects with normal weight were slightly older than those in the other two groups (76.3 years vs. 75.1 and 73.1 years in overweight and obese groups, respectively; p = 0.033).

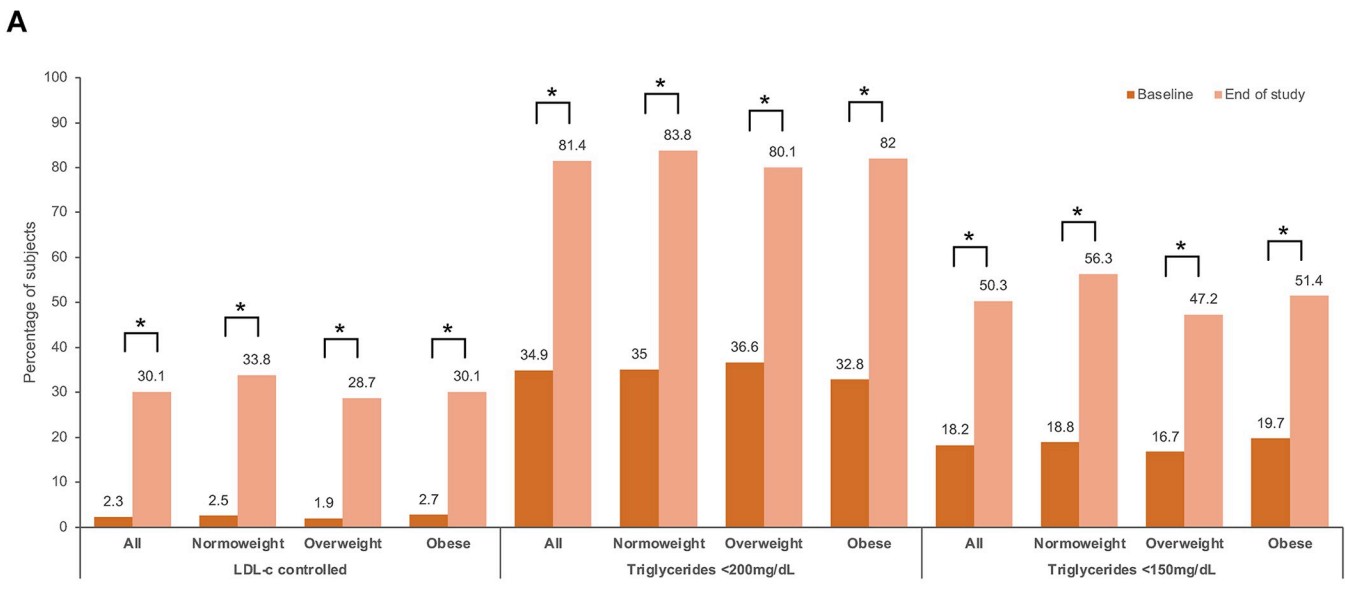

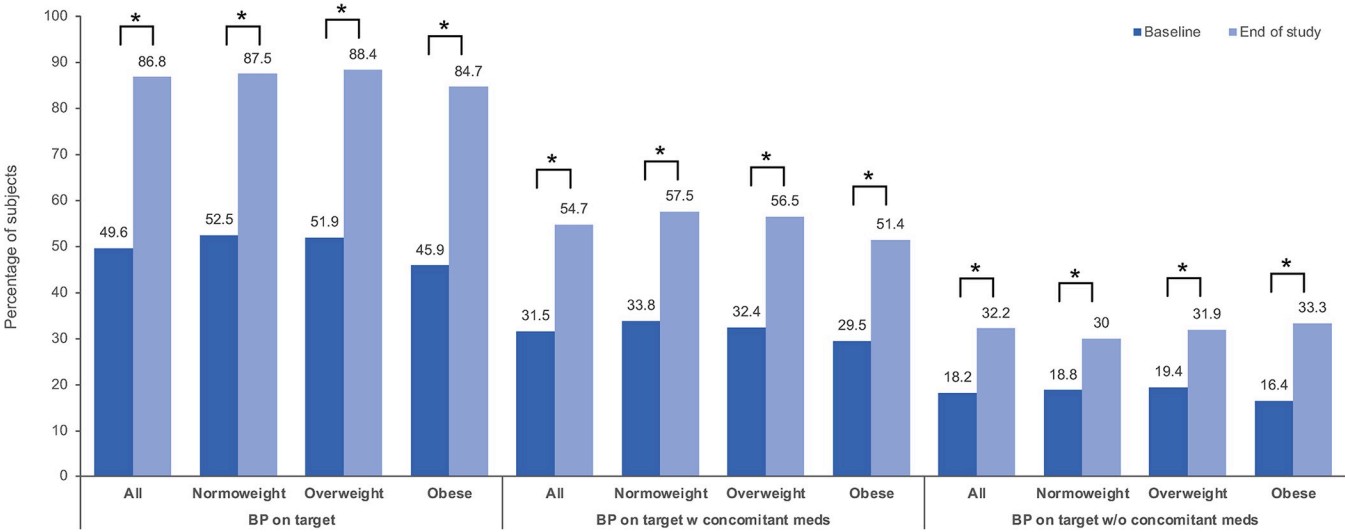

**Fig 3.** Achievement of target LDL-c levels and recommended triglyceride levels (A) and blood pressure (B) with the CV polypill compared to baseline. ***P<0.0001; **P<0.01; *P<0.05. BP: blood pressure; LDL-c: low-density lipoprotein cholesterol; w: with; w/o: without.

Similar to what was observed in the overall population, all assessed lipid parameters improved compared to baseline regardless of BMI category after initiating the CV polypill treatment (**S4 Fig**). Moreover, the magnitude of the observed change with respect to baseline (%) was comparable in all cases to those observed in the overall population, with decreases ≥30% for TC, TG, and LDL-c, and increases ≥10% for HDL-c. Regarding BP values, there was a significant decrease in all BMI groups compared to baseline ($p<0.001$), and the magnitude of the reduction was between 11% and 13% for SBP and between 9% and 12% for DBP across weight groups (**S4 Fig**).

The proportion of elderly patients with LDL-c controlled at baseline was as low as in the overall population (3.1% and 2.3%, respectively) and increased significantly up to 33.8% (30.1 to 33.8% across BMI groups; all $p<0.001$) at the end of the study (**S5 Fig**). While less than half of the elderly patients had TG levels <200 mg/dL (43.3% overall) and about one third <150 mg/dL (24.1% overall) at baseline, these proportions increased significantly after CV polypill treatment in all BMI categories (up to >85% and >48% across groups, respectively, all $p<0.001$). Regarding BP, less than half of the elderly patients were at recommended levels at baseline (42.4% overall) but this proportion significantly increased up to 76.8% after treatment with the CV polypill (75.3% to 80.0% across BMI categories; all $p<0.001$) (**S5 Fig**). Finally, at the end of the study, 60.7% of elderly patients needed concomitant antihypertensives to attain BP control and 16.1% had BP on target with the CV polypill as a single treatment ($p<0.001$ and $p = 0.008$, respectively).

## Adverse effects during the treatment with the CV polypill

The proportion of patients spontaneously reporting adverse events during follow-up was 9.1%. Except cough (5.1% of incident cases), the frequency of all reported adverse events was <2.5% (gastritis/gastric ulcer and upper gastrointestinal bleeding 0.6%; severe left ventricular congestive heart failure 0.4%, acetylsalicylic acid intolerance 0.2%, and other 2.3%).

## Discussion

In this study, control of CV risk factors (i.e., blood pressure and lipid profile) in individuals with established CVD improved with CV polypill treatment regardless of the initial patient BMI. Moreover, this improvement was comparable to that observed in the elderly subpopulation.

The overall profile of the studied population showed that patients were middle aged, more than 80% overweight or obese, and more than half of them with concurrent risk factors regardless of initial weight. This picture reflects the general Mexican population, where 73% of subjects are overweight and the prevalence of abnormal lipid profiles among this group is particularly high (about 33–35% have hypertriglyceridemia and 18% mixed dyslipidemia) [3, 32]. However, it is striking that CV risk factor control in our population, that is, patients in secondary prevention, was rather poor. For instance, 61% of them were current smokers, 60% had hypertension, 82% hypertriglyceridemia, and the mean LDL-c levels were far above the recommended target. Of note, nearly 50% of our overall population was not treated with statins or other secondary prevention drugs, which is consistent with the figures available in a recent study conducted in 41 low and middle-income countries, where less than 40% of patients in secondary prevention in Mexico was reported to be treated with statins [33]. It is also in line with another recent study conducted in the Mexican population where only 60% of patients in secondary prevention were on high-intensity statin treatment [34].

After treatment with the CV polypill, the mean levels of lipid parameters (TC, TG, and LDL-c) decreased 31% to 43% across all BMI groups, the mean HDL-c levels increased between 13% and 14%, and the lipid ratios, indicative of atherogenic dyslipidemia, almost halved (i.e., TC/HD-c and LDL-c/HDL-c decreased between –39% and –50% across groups). This is in line with the 25–30% reductions in TC, TG, and LDL-c reported by prospective, hospital-based studies conducted in the Mexican population, assessing the effectiveness of the CV polypill after 6 months in patients with myocardial infarction [35, 36]. Our findings also agree with the SORS observational and prospective registry study conducted in Mexico, including patients in secondary prevention (22–23% reduction in TC and LDL after 12 months) [37]. Finally, the LDL-c reduction was higher in our study than in the retrospective Spanish

NEPTUNO study (15.4% reduction after 2 years) [25], and also higher than the European SECURE randomized trial (10.3% reduction after 3 years) [24], both evaluating patients with established CVD. The improvement observed with the CV polypill cannot be possibly attributed to prior treatment with less potent statins in our study, as more than 90% of the patients were on equipotent or more potent statins before initiating the CV polypill strategy. It has been recently suggested that LDL-c reduction with the CV polypill would be equivalent to doubling the dose of a statin in monotherapy because of a potential pharmacodynamic interaction between the components (acetylsalicylic acid, atorvastatin, and ramipril) [38]. Although the statin in the CV polypill that we used was simvastatin, this synergistic effect could potentially explain the LDL-c improvement even in patients previously on more potent statins [39]. Additionally, CV polypill use substantially increased the proportion of patients with TG levels below 200 mg/dL at the end of the study to 80%, but the proportion of patients at LDL-c target was still low (about 30% of patients). The latter could be attributed to the low observed use of additional drugs on top of the statin component of the CV polypill (0.8% of cases) when patients did not achieve lipid control, which is in line with the 8.3% of patients prescribed additional lipid-lowering drugs reported by a polypill study conducted in Australia [40]. It is not uncommon that patients require additional lipid-lowering drugs to achieve lipid targets [30, 41], and the CV polypill strategy must be seen as a baseline treatment able to control different risk factors that offers the flexibility to personalize the treatment through the addition of other drugs when the goals are not met.

Regarding BP control, we observed a 10–11% decrease in SBP and 5–6% in DBP after polypill treatment, with no differences between weight groups, such that more than 80% of patients were at recommended levels at the end of the study. These results align with those reported by the SORS hospital-based observational, prospective Mexican study including patients in secondary prevention (11.5% and 10% reduction in SBP and DBP, respectively) [37]. They also align with the reductions reported in the retrospective NEPTUNO study conducted in patients with established CVD (10.1% and 5.9% reduction in SBP and DBP, respectively) [25]. Moreover, 55% of patients at target BP used concomitant anti-hypertensive medication to remain controlled in addition to the baseline CV polypill. This also aligns with another study reporting that 62% of patients still need additional BP treatment on top of the CV polypill [40]. Nevertheless, we observed that the proportion of subjects with BP on target increased compared to prior treatment in both groups, patients needing and not needing concomitant medication. This result demonstrates that regardless of the prior antihypertensive treatment, the CV polypill increased the overall proportion of controlled patients.

Vascular or heart age is an emerging health metric in primary care defined as the CVD risk of an individual transformed to a person's age if the same risk factors were optimally controlled [42]. This allows translating the abstract concept of 10-year CV risk into quite an understandable message for the patient, leading to more significant decreases in CV risk factors than the standard of care [43]. For instance, it would inform a 40-year-old male smoker with hypertension and hypercholesterolemia (moderate risk as per SCORE scale) that he has a vascular age comparable to a healthy 63-year-old man [43]. In our study, we observed a reduction of 6.6 years in the cardiovascular age of normal weight patients and 5.2 and 5.0 years in the overweight and obese groups, respectively. The fact that the CV polypill could exert more intense protective effects on CVD risk than the multi-pill strategy in patients on secondary prevention was originally a hypothesis [35, 37, 44]. However, a very recent retrospective study (NEPTUNO) demonstrated that, after 2 years of treatment, the incidence of recurrent major cardiovascular events in patients in secondary prevention was lower among those treated with the CV polypill than in matched patients taking the individual components separately, on equipotent drugs, or on other therapies [25]. More recently, the prospective, randomized

SECURE study showed that in 2500 patients >65 years who have recently had a myocardial infarction (MI) with a mean follow-up of 3 years, the polypill strategy reduced the relative risk of recurrent major adverse CV events (MACE: MI, stroke, revascularization, and CV death) by 24% and the CV mortality by 33% compared with the usual care arm [24].

The economic costs of implementing the CV polypill strategy were out of the scope of this study, but the price of the polypill is generally perceived as a barrier in low/medium socioeconomic populations with limited financial resources in the health system [45]. However, a recent meta-analysis of socioeconomic studies concluded that polypills resulted in improved adherence and quality of life at a price equal to or lower than multiple monotherapies, with the price (which was below the commonly accepted thresholds or cost saving in both primary and secondary prevention) as one of the key drivers of the cost-effectiveness [16]. Similarly, a cost-utility analysis conducted in Portugal documented that the CV polypill was consistently cost-effective compared with monocomponents in secondary prevention [17].

One weakness of the study is that the sample size was relatively small, thus with low statistical power potentially resulting in a type II error limiting the generalizability of the results to larger populations of patients. There are also several limitations inherent to the retrospective design that must be acknowledged. Firstly, data were extracted from hospital medical records, and it is possible that not all pertinent risk factors or predictor variables were identified or subsequently recorded. For instance, several meta-analyses have shown that the CV polypill strategy has a significant positive impact on medication adherence, with more than 76% of patients showing medication adherence [46–48]. This was also documented in trials using the same CV polypill as in our study in patients in secondary prevention. For instance, in the randomized FOCUS trial [49], the adherence rate for the CV polypill was 65.7% at 9 months vs 55.7% in patients taking the three drugs separately. In the recent SECURE trial [24], 74.1% of the patients in the polypill group and 63.2% of those in the usual-care group were adherent after 24 months of follow-up. In the present research, medication adherence was not included in the analyses because it was self-reported by the patient when asked by the treating physician and documented as unstructured free text in the medical records. However, adherence rates were not expected to differ between weight groups because all patients were treated with the CV polypill. Moreover, we cannot discard inaccuracies in the recorded data on laboratory parameters. Additionally, the fact that different healthcare professionals attending the Coronary Unit were involved in patient care during follow-up implies that the measurements (i.e., use of concomitant medications) could be less accurate than those otherwise obtained through a prospective cohort study design. Moreover, the study had a non-randomized design without a control group (case series) prone to selection bias. Indeed, the decision to initiate or switch to the CV polypill strategy was made by the treating physician (and the patient) while providing usual care. As such, the results of the study may be susceptible to selection bias favoring patients for whom the CV polypill was best suited (e.g., poorly adherent) according to the preferences or experience of individual doctors or local protocols. This bias could have been partially lessened by the retrospective design since the data were collected before the research question was formulated [50]. Additionally, because of the retrospective design and lack of a control group, we can only establish an association and not a cause-effect between the CV polypill strategy and the observed improvement in risk factors. Besides, only about half of the patients were treated with RAAS inhibitors or statins before initiating the CV polypill strategy, which could have resulted in an overestimation of its effect on the studied outcomes. Finally, we cannot actually distinguish whether the improvement in risk factor control was due to the CV polypill strategy or to other drugs used in combination. However, the proportion of patients requiring different drug classes as concomitant medications to optimize LDL-c and BP control was similar before initiating (or switching to) the CV polypill, at the time it was

started and at the end of the follow-up. Thus, it's not likely that drugs added to the baseline CV polypill treatment were responsible for the observed improvement.

## Conclusions

To best of our knowledge, this is the first study assessing the impact of the CV polypill as a secondary prevention strategy as a function of weight stratification. In the Mexican population, where hypertension and hypercholesterolemia are prevalent and the treatment rates are low, the current findings add to a growing body of literature on the use of the CV polypill as baseline therapy, providing evidence that it is a beneficial strategy regardless of the patient's BMI. The fact that it was safe and effective across subjects with distinct metabolic and CVD risk profiles has important implications for future practice, particularly in countries where obesity and overweight are a significant public health issue and there is an urgent need to improve the CV risk factor control and decrease the associated excess CV risk and incidence of CV events.

For all these reasons, the use of the CV polypill as baseline therapy for secondary prevention seems to be a reasonable strategy that enhances CV risk factor control regardless of the patient's BMI.

## Supporting information

**S1 Fig. Overall proportion of patients receiving ASA, RAAS inhibitors and statins A) prior to initiating or switching to the CV polypill strategy, at baseline, and at the end of follow-up and B) by BMI group prior to CV polypill initiation.** ASA: acetylsalicylic acid; RAAS: renin-angiotensin-aldosterone system.
(PDF)

**S2 Fig. Proportion of patients by BMI group receiving equipotent, more potent, and less potent statins (A) or RAAS inhibitor (B) prior to initiating or switching to the CV polypill strategy.**
(PDF)

**S3 Fig. Evolution of concomitant medications in the overall population during the different assessed periods.** ASA: acetylsalicylic acid; BB: beta blockers; DIU: diuretics; CCB: calcium channel blockers.
(PDF)

**S4 Fig. Change in levels and additional change (%) in lipid parameters and blood pressure with the CV polypill compared to baseline in patients >65 years.** ***P<0.001 in all weight groups. DBP: diastolic blood pressure; HDL-c: high-density lipoprotein cholesterol; LDL-c: low-density lipoprotein cholesterol; SBP: systolic blood pressure; TC: total cholesterol; TG: triglycerides.
(PDF)

**S5 Fig. Achievement of target levels of LDL-c and recommended levels of triglycerides (A) and blood pressure (B) with the CV polypill compared to baseline in patients >65 years.** ***P<0.0001; **P<0.01; *P<0.05. BP: blood pressure; LDL-c: low-density lipoprotein cholesterol; w: with; w/o: without.
(PDF)

**S1 Table. Reasons for CV polypill discontinuation in the overall population and by BMI subgroup.**
(PDF)

**S2 Table. Evolution of concomitant medications during the different assessed periods by BMI group.** BMI: body mass index; DBP: diastolic blood pressure; HDL-c: high-density lipoprotein cholesterol; LDL-c; low-density lipoprotein cholesterol; SBP: systolic blood pressure; SD: standard deviation; TC: total cholesterol; TG: triglycerides.
(PDF)

**S3 Table. Baseline demographic and clinical characteristics of patients >65 years.** BMI: body mass index; DBP: diastolic blood pressure; HDL-c: high-density lipoprotein cholesterol; LDL-c; low-density lipoprotein cholesterol; SBP: systolic blood pressure; SD: standard deviation; TC: total cholesterol; TG: triglycerides.
(PDF)

## Author Contributions

**Conceptualization:** José Alejandro Chávez Fernández, Marcelo Ramírez Mendoza, Hermelinda Kassck Ipinaa, Luís Antonio Sánchez Ángeles, Galileo Escobedo, Lucía Angélica Méndez-García.

**Formal analysis:** José Alejandro Chávez Fernández.

**Funding acquisition:** José Alejandro Chávez Fernández.

**Investigation:** José Alejandro Chávez Fernández, Marcelo Ramírez Mendoza, Hermelinda Kassck Ipinaa, Luís Antonio Sánchez Ángeles, Antonio González Chávez, Galileo Escobedo, Lucía Angélica Méndez-García.

**Methodology:** José Alejandro Chávez Fernández, Marcelo Ramírez Mendoza, Hermelinda Kassck Ipinaa, Luís Antonio Sánchez Ángeles, Antonio González Chávez, Galileo Escobedo, Lucía Angélica Méndez-García.

**Project administration:** José Alejandro Chávez Fernández.

**Resources:** José Alejandro Chávez Fernández.

**Supervision:** José Alejandro Chávez Fernández.

**Writing – original draft:** José Alejandro Chávez Fernández.

**Writing – review & editing:** José Alejandro Chávez Fernández, Marcelo Ramírez Mendoza, Hermelinda Kassck Ipinaa, Luís Antonio Sánchez Ángeles, Antonio González Chávez, Galileo Escobedo, Lucía Angélica Méndez-García.

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
