## [Decision Letter · Decision Letter 0]

10 May 2023

PONE-D-23-04800The cardiovascular polypill as baseline treatment improves lipid profile and blood pressure regardless of body mass index in patients with cardiovascular disease. The Bacus studyPLOS ONE

Dear Dr. Chavez fernandez,

Thank you for submitting your manuscript to PLOS ONE. After careful consideration, we feel that it has merit but does not fully meet PLOS ONE’s publication criteria as it currently stands. Therefore, we invite you to submit a revised version of the manuscript that addresses the points raised during the review process.

We look forward to receiving your revised manuscript.

Kind regards,

Kamal Sharma

Academic Editor

PLOS ONE

Journal Requirements:

“Medical writing and editorial support were provided by Mònica Gratacòs funded by Ferrer Internacional SA.”

“The study was partly funded by Ferrer Internacional SA. The funder had no role in study design, data collection, data interpretation, or decision to submit it for publication.”

“JAC received payment from Ferrer Internacional SA to conduct the data analysis and prepare the manuscript. The other authors report no relationships that could be construed as a conflict of interest.”

Additional Editor Comments:

Hello,

It is an interesting study that requires major revision.

We have received 4 reviews for the study. 3 have recommended minor revision while 1 has suggested rejection.

It is worthwhile to review the article after the major revision addressing all the recommendations if possible of all the 4 reviewers.

thanks

Reviewers' comments:

Reviewer's Responses to Questions

**Comments to the Author**

1. Is the manuscript technically sound, and do the data support the conclusions?

Reviewer #1: Yes

Reviewer #2: No

Reviewer #3: Yes

Reviewer #4: Yes

2. Has the statistical analysis been performed appropriately and rigorously? 

Reviewer #1: I Don't Know

Reviewer #2: Yes

Reviewer #3: Yes

Reviewer #4: Yes

3. Have the authors made all data underlying the findings in their manuscript fully available?

Reviewer #1: Yes

Reviewer #2: Yes

Reviewer #3: Yes

Reviewer #4: Yes

4. Is the manuscript presented in an intelligible fashion and written in standard English?

Reviewer #1: Yes

Reviewer #2: Yes

Reviewer #3: Yes

Reviewer #4: Yes

5. Review Comments to the Author

Reviewer #1: Fernandez et al. demonstrated beneficial effects of polypill on lipid profile and blood pressure in patients with cardiovascular disease (CVD) in Mexico. Polypill improved lipid profile and decreased blood pressure and vascular age regardless body mass index (BMI) in the Mexican population, who had high prevalence rate of hypertension and hypercholesterolemia. The effects of polypill for secondary prevention of CVD have already reported in several studies. However, this study is unique in analyzing the effects of polypill by BMI stratification. There are several concerns which should be addressed.

1. This study is a single center retrospective one. The authors should describe about patient selection in more detail. Were consecutive cases enrolled?

2. In this study, patients with established coronary heart disease were enrolled. Please clearly define ‘established coronary heart disease’ in Methods section.

3. Are there any data about cardiovascular events or death during follow-up periods? The information would be useful for understanding the clinical efficacy of polypill on secondary prevention.

4. Is there a difference in lipid profile and blood pressure between presence and absence of prior prescription of statin or RAS BI to CV polypill?

5. In Figure 1 and Figure 2, Y-axis scale should start from zero.

6. The authors should explain S2 Fig in Results section.

7. Detailed explanation about vascular age would be helpful for understanding its significance.

8. The mean time treatment with the CV polypill and follow-up period should be shown in each subgroup in Table 1.

9. What kind of statistical test was performed in this study?

Reviewer #2: Authors report an observational, retrospective study reviewing the hospital medical records

of 479 patients with CV disease who initiated treatment with the CV polypill

between 2013 and 2019 at a general hospital in Mexico. Patients were grouped as

normal weight, overweight or obese according to their initial body mass index (BMI).

They collected blood pressure, lipid profile, and vascular age at the last visit

recorded during the period following treatment.

They report that lipid parameters and blood pressure improved compared to baseline

regardless of the initial BMI category. Polypill resulted a significant overall improvement in lipid and BP

control. As authors addressed the study comes with significant limitations.

1. The study is retrospective design.

2. Data were extracted from hospital medical records, and methodological bias cannot be excluded.

3. There is no control group in the study.

4. Authors state that inaccuracies in data collection due to the design cannot be excluded.

5. Selection bias is likely for polypill that favoring patients for polypill will be the ones with poor compliance.

6. No study protocol is present due to the design.

7. Patients were not treated adequately prior to the polypill.

8. The study has limited novelty. Several polypills have been studied in numerous research trials in comparison to placebo or usual care.

9. Previous clinical trials have measured adherence rates, adverse events and risk factor control.

10. This trial design would not have the power to test the impact of polypill on clinical outcomes.

11. Use of polypill will come with concerns i.e. that a side-effect of any one of the components can lead to discontinuation of all drugs.

12. Clinicians by training are taught to titrate dosages to ‘treat to target’ various risk factors. The key worry for most practitioners has been the inconvenience of dose adjustments with a polypill. The lack of ability to fine tune one component of the polypill if the therapeutic goals are not reached is a constant concern.

13. Drug interractions are of concern. I.e. conflicting reports exist on the concomitant use of aspirin and ACE inhibitors.

Reviewer #3: You have made a good attempt at answering the question whether polypill improves lipid profile and BP regardless of BMI, inpatients with CAD. In my discussion I have asked for clarifications and suggested a focused discussion that highlights your findings. I have also suggested that your findings be compared with degree of BP and lipid lowering in light of previous studies as outlined in my review. All these are minor adjustments.

Reviewer #4: A well written manuscript.

a) May I suggest using the word 'control' in the limitation section. As you rightly pointed out, you cannot prove cause-effect here as the study was a retrospective study design, with undoubtedly no control population.

b) Would also go into costings here, as this may also be an issue in widespread implementation of such a therapy.

3) Can you also go into hospitalisation/revascularisation/deaths? Is the proportion of participants who succumbed to this outcomes similar to statistics in the preceding years? Any differences compared to the ones who discontinued the polypill?

6. PLOS authors have the option to publish the peer review history of their article (what does this mean?). If published, this will include your full peer review and any attached files.

Reviewer #1: No

Reviewer #2: **Yes: **Mehmet Agirbasli MD

Reviewer #3: No

Reviewer #4: **Yes: **Dr Mark Abela

---

## [Author Response · Author response to Decision Letter 0]

29 Jun 2023

A rebuttal letter that responds to each point raised by reviewershas been uploaded as a separate file labeled 'Response to Reviewers'.

Comments by the Editor have been responded in the cover letter.

---

## [Decision Letter · Decision Letter 1]

11 Aug 2023

PONE-D-23-04800R1

The cardiovascular polypill as baseline treatment improves lipid profile and blood pressure regardless of body mass index in patients with cardiovascular disease. The Bacus study

PLOS ONE

Dear Dr. Chavez fernandez,

Thank you for submitting your manuscript to PLOS ONE. After careful consideration, we feel that it has merit but does not fully meet PLOS ONE’s publication criteria as it currently stands. Therefore, we invite you to submit a revised version of the manuscript that addresses the points raised during the review process.

ACADEMIC EDITOR:

It is a nice paper and needs minor changes to incorporate limitations of the study being a retrospective single centre data set and assigned to look at soft end point and hence adding the need for a larger multicentric prospective interventional outcome study may be more helpful in future. 

We look forward to receiving your revised manuscript.

Kind regards,

Kamal Sharma

Academic Editor

PLOS ONE
---

## [Editor Report · Acceptance letter]

17 Aug 2023

PONE-D-23-04800R1 

The cardiovascular polypill as baseline treatment improves lipid profile and blood pressure regardless of body mass index in patients with cardiovascular disease. The Bacus study. 

Dear Dr. Chávez Fernández:

I'm pleased to inform you that your manuscript has been deemed suitable for publication in PLOS ONE. Congratulations! Your manuscript is now with our production department. 

Kind regards, 

on behalf of

Dr. Kamal Sharma 

Academic Editor

PLOS ONE